# Post-transcriptional splicing can occur in a slow-moving zone around the gene

Allison Coté[1†], Aoife O'Farrell[1†], Ian Dardani[1], Margaret Dunagin[1], Chris Coté[1], Yihan Wan[2], Sareh Bayatpour[1], Heather L Drexler[3], Katherine A Alexander[4], Fei Chen[5], Asmamaw T Wassie[6], Rohan Patel[1], Kenneth Pham[6], Edward S Boyden[7], Shelly Berger[4], Jennifer Phillips-Cremins[1], L Stirling Churchman[3], Arjun Raj[1,8]*

[1]Department of Bioengineering, University of Pennsylvania, Philadelphia, United States; [2]School of Life Sciences, Westlake University, Hangzhou, China; [3]Department of Genetics, Blavatnik Institute, Harvard Medical School, Boston, United States; [4]Department of Cell and Developmental Biology, Penn Institute of Epigenetics, Perelman School of Medicine, University of Pennsylvania, Philadelphia, United States; [5]Broad Institute of MIT and Harvard, Cambridge, United States; [6]Department of Cell and Molecular Biology, University of Pennsylvania, Philadelphia, United States; [7]Departments of Biological Engineering and Brain and Cognitive Sciences, Media Lab and McGovern Institute, Massachusetts Institute of Technology, Cambridge, United States; [8]Department of Genetics, Perelman School of Medicine, University of Pennsylvania, Philadelphia, United States

*For correspondence:
arjunraj@seas.upenn.edu

[†]These authors contributed equally to this work

Competing interest: The authors declare that no competing interests exist.

**Abstract** Splicing is the stepwise molecular process by which introns are removed from pre-mRNA and exons are joined together to form mature mRNA sequences. The ordering and spatial distribution of these steps remain controversial, with opposing models suggesting splicing occurs either during or after transcription. We used single-molecule RNA FISH, expansion microscopy, and live-cell imaging to reveal the spatiotemporal distribution of nascent transcripts in mammalian cells. At super-resolution levels, we found that pre-mRNA formed clouds around the transcription site. These clouds indicate the existence of a transcription-site-proximal zone through which RNA move more slowly than in the nucleoplasm. Full-length pre-mRNA undergo continuous splicing as they move through this zone following transcription, suggesting a model in which splicing can occur post-transcriptionally but still within the proximity of the transcription site, thus seeming co-transcriptional by most assays. These results may unify conflicting reports of co-transcriptional versus post-transcriptional splicing.

## eLife assessment

This **fundamental** study addresses a long-standing mystery in splicing regulation: does splicing occur co- or post-transcriptionally? The authors provide **compelling** evidence demonstrating that splicing can occur post-transcriptionally at a transcription site proximal zone, changing the way we think about splicing.

## Introduction

The mRNA transcribed by many eukaryotic genes are spliced, a process in which the intronic RNA are removed and the exonic RNA are joined together to form the ultimate mature mRNA. A major question in the field is how tightly associated the processes of transcription and splicing are. Some work suggests that splicing occurs very shortly after the RNA polymerase transcribes a particular splice

junction (*Beyer et al., 1981*; *Oesterreich et al., 2016*; *Khodor et al., 2011*; *Reimer et al., 2021*), while other work suggests that many pre-mRNA are fully transcribed before splicing occurs (*Tsai et al., 1980*; *Drexler et al., 2020*; *Coulon et al., 2014*; *Khodor et al., 2012*; *Choquet et al., 2022*). The relative spatial locations of nascent pre-mRNA, fully transcribed pre-mRNA, and mature mRNA species have the potential to directly reveal where—and consequently in what order—the processes of transcription and splicing occur. However, to date, the use of molecular imaging to systematically measure the locations of these partially processed RNA intermediates has been limited in scope, interrogating either single-intron reporter genes or single introns within endogenous genes (*Coulon et al., 2014*; *Waks et al., 2011*; *Vargas et al., 2011*).

In lieu of direct visualization, many studies have used biochemical fractionation to infer the location of various intermediates (*Drexler et al., 2020*; *Tilgner et al., 2012*; *Wuarin and Schibler, 1994*; *Bhatt et al., 2012*; *Pandya-Jones et al., 2013*; *Pandya-Jones and Black, 2009*; *Mayer et al., 2015*). Fractionation methods separate cellular RNA into different compartments, such as the putatively chromatin-associated RNA, nucleoplasmic RNA, and cytoplasmic RNA (*Wuarin and Schibler, 1994*; *Mayer and Churchman, 2017*). The implicit assumption made by such fractionation-based methods is that the RNA species in the 'chromatin fraction' represent nascent pre-mRNA that are tethered to the gene body by the RNA polymerase II itself, and that once the pre-mRNA disengages with RNA polymerase II, it immediately moves directly into the nucleoplasm. Under these assumptions, any splicing observed in the chromatin fraction would be assumed to be co-transcriptional. However, this assumption may not hold; it is possible that pre-mRNA remains in a chromatin-associated compartment for some time after transcription completes, and thus splicing observed in the chromatin compartment may in fact still be post-transcriptional (*Brody et al., 2011*). Some groups have further purified nascent RNA via metabolic labeling or by using RNA polymerase II antibodies, but these methods still have the potential to co-purify mature RNA (*Nojima et al., 2015*). Ultimately, such alternative explanations are difficult to eliminate without an independent and explicit verification of which RNA intermediates reside in particular compartments.

Advances in RNA imaging have enabled researchers to image RNA intermediates with single-molecule resolution, both in fixed and in living cells (*Coulon et al., 2014*; *Waks et al., 2011*; *Vargas et al., 2011*; *Zhang et al., 1994*; *Martin et al., 2013*; *Levesque and Raj, 2013*). Imaging using probes targeting both exonic and intronic regions of RNA has revealed bright nuclear foci that represent nascently transcribing RNA (*Vargas et al., 2011*; *Levesque and Raj, 2013*). The general lack of intronic signal away from these transcription sites has been taken as evidence for co-transcriptional splicing, with notable examples of post-transcriptional splicing at speckles being observed in special cases (*Vargas et al., 2011*). However, owing to the diffraction limit for optical microscopy, it has been difficult to visualize RNA intermediates in the immediate vicinity of the gene undergoing transcription, thus making it difficult to observe whether RNA are still actively being transcribed during splicing or remain at the site of transcription for some time after transcription is complete. Since splicing would appear to occur in the 'chromatin fraction' in both of these scenarios, it is possible that much of splicing actually occurs in this transcription-proximal region after transcription is complete ('proximal post-transcriptional splicing'). Indeed, recent live-cell imaging methods showed that the splicing of a reporter gene is 85% post-transcriptional (*Coulon et al., 2014*), suggesting the latter possibility, but as RNA from endogenous genes may be processed differently, the use of reporter genes leaves open the question of when endogenous genes undergo splicing relative to their transcription.

Here, we designed probes to comprehensively interrogate the spatial localization of several RNA intermediates using a combination of RNA FISH and expansion microscopy (*Chen et al., 2016*; *Chen et al., 2015*), allowing for visualization of newly transcribed RNA at high resolution. We found that the proportion of splicing that occurs post-transcriptionally varies from intron to intron within a single gene, but that all endogenous genes we tested displayed at least some degree of post-transcriptional splicing. We also employed expansion microscopy and live-cell imaging to demonstrate that newly synthesized RNA dwell and undergo continuous splicing near the site of transcription after transcription is complete within a proximal slow-moving zone. These RNA are untethered to the site of transcription and eventually diffuse into either the nucleoplasm or near nuclear speckles. These results suggest a model for splicing dynamics that unifies existing data.

## Results

### At least one intron of each observed endogenous gene is spliced post-transcriptionally

The extent to which splicing is coupled to transcription, both in space (distance from transcription site) and in time (time since transcription has begun), has been difficult to measure. To address this question, we directly visualized the locations of spliced and unspliced RNA relative to the site of transcription in situ using single-molecule RNA FISH (*Raj et al., 2008*; *Femino et al., 1998*), to simultaneously fluorescently label the exons and several individual introns for a number of genes of interest: *CPS1*, *EEF2*, *TM4SF1*, and *FKBP5*. *CPS1* was chosen because it is a long and highly expressed gene. *EEF2* was chosen because it is a highly expressed housekeeping gene. *TM4SF1* was chosen because it is highly expressed and sequencing data suggested it was highly post-transcriptionally spliced (see Materials and methods for details). Finally, *FKBP5* was chosen because it is inducible in A549 cells by application of dexamethasone.

By distinguishing the separate fluorescent signals from probes bound to exons and introns, we could visualize splicing intermediates (represented by colocalized intron and exon spots) relative to the site of transcription (represented by bright colocalized intron and exon spots) and fully spliced products (represented by exon spots alone). We were particularly interested in visualizing the location of splicing intermediates in order to distinguish between two possibilities: (1) an intron from a pre-mRNA being spliced out <u>at</u> the site of transcription and (2) an intron from a pre-mRNA being spliced out <u>away from</u> the site of transcription (*Figure 1A*; right; i and ii versus iii). Observing spliced RNA at the site of transcription could represent RNA that is spliced either co-transcriptionally (tracking along with or shortly behind the polymerase; *Figure 1Ai*) or proximally post-transcriptionally (near the site of transcription but after the process of transcription has been completed) (*Figure 1Aii*). However, observing pre-mRNA with unspliced introns that were sufficiently far away from the site of transcription such that they could no longer be attached to the location of the gene itself would suggest that that intron must be spliced out post-transcriptionally (distal post-transcriptional splicing, *Figure 1Aiii*).

To determine what fraction of splicing occurred far from the site of transcription, we first needed to classify each colocalized exon and intron spot as either a transcription site or a dispersed pre-mRNA. We computationally identified spots for both introns and exons of a particular gene, then each intron spot that was within 0.65 µm of an exon spot was designated a colocalized exon and intron spot (0.65 µm is the 75th percentile of mRNA lengths determined by measuring the distance from 5' to 3' signal; *Figure 1B*, see *Figure 1—figure supplement 2C* for mRNA lengths), which we assumed represents at least one nascent pre-mRNA (potentially several nascent pre-mRNA when found at the site of transcription). We chose this colocalization threshold such that at least 78% of exons (depending on the gene, *CPS1*=78%, *EEF2*=85%, *TM4SF1*=87%) colocalized with other exons of the same gene in the cytoplasm; however, varying this threshold from 0.13 µm (one pixel) to 2.6 µm (20 pixels) did not significantly change dispersal patterns (*Figure 1—figure supplement 2A*). Intron spots that did not colocalize with an exon spot were presumed to be degradation products or non-specific background and were discarded (these were generally <25% of intron spots; *Supplementary file 1A*). We used an intensity threshold to categorize each colocalized exon and intron spot as either a transcription site or a dispersed pre-mRNA (*Figure 1*, see *Figure 1—figure supplement 1A and B* for the scheme for the classification of transcription sites). We also tried several other methods for choosing transcription sites, all of which showed qualitatively similar results (*Figure 1—figure supplement 1A*), showing that the detection of dispersal is not dependent on the choice of method. Our chosen strategy likely grouped together the signal from multiple pre-mRNA (introns and exons) at the transcription site, rendering them indistinguishable at this level of spatial resolution; we separately analyzed these transcription sites further later (see *Figure 2*).

We then calculated the distance of each dispersed pre-mRNA from the nearest transcription site for all cells, yielding population-wide measurements of dispersal per intron (*Figure 1D*). We found that for the genes *CPS1*, *EEF2*, *TM4SF1*, and *FKBP5*, at least one intron was present in dispersed pre-mRNA in both transformed cell lines (HeLa and A549) and non-transformed cells (CRL-2097) (three to five introns tested per gene, *Figure 1D*), suggesting that pre-mRNA dispersal is a generic feature of transcription and splicing that is not dependent on the particular cell line used. However, we found that the dispersal patterns for any one particular intron were not necessarily consistent between HeLa and CRL-2097 cells, suggesting splicing patterns may be influenced by cellular context. The presence

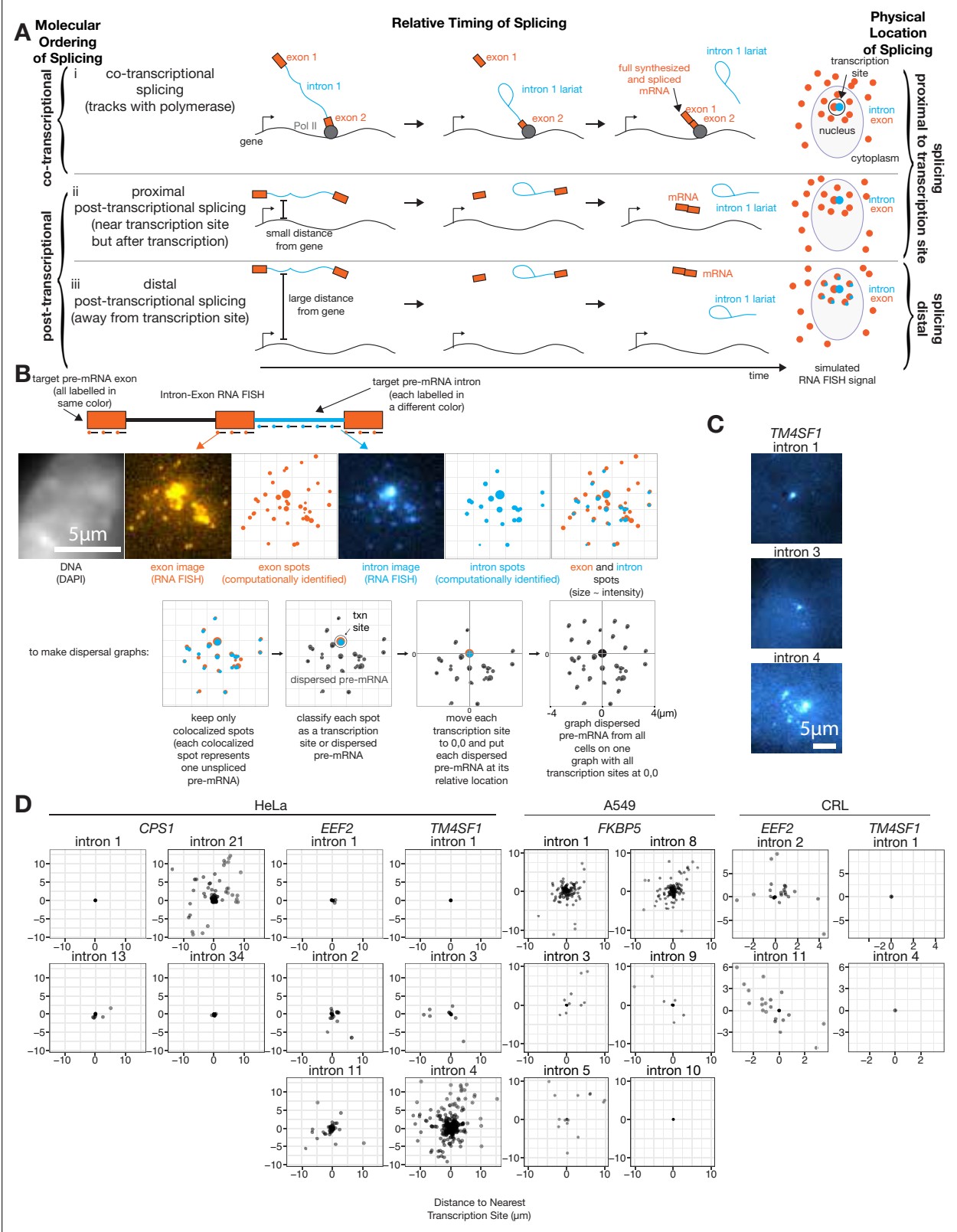

**Figure 1.** At least one intron from every tested endogenous gene is post-transcriptionally spliced. (**A**) Schematic depicting three categories of splicing. (**B**) Schematic depicting RNA FISH method and translation of RNA FISH images into dispersal graphs. Computed spots are calculated by fitting a Gaussian to the distribution of intensity signals and pinpointing the center of that Gaussian at sub-pixel resolution. Gene depicted is *FKBP5*. (**C**) Example images of dispersed (intron 4) and not dispersed (introns 1 and 3) transcription sites, via RNA FISH. Each image is a max merge of 30 optical

*Figure 1 continued on next page*

*Figure 1 continued*

z-sections, 0.3 µm step distance. (**D**) Graphs indicating dispersal distance or the distance of each detected pre-mRNA to its nearest transcription site. Leftmost graphs of CPS1, EEF2, and TM4SF1 data are from HeLa cells, FKBP5 data is from A549 cells after 8 hr in dexamethasone, rightmost graphs of EEF2 and TM4SF1 data are from CRL-2097. All scale bars represent 5 µm unless otherwise noted.

The online version of this article includes the following figure supplement(s) for figure 1:

**Figure supplement 1.** Transcription site choice.

**Figure supplement 2.** Defining post-transcriptionality.

**Figure supplement 3.** Sequencing corroborates RNA FISH dispersal results.

of intermediates away from the site of transcription showed that for all the genes we tested, some introns were spliced post-transcriptionally and away from the gene body itself. We never observed introns in cytoplasmic mRNA, suggesting that all of these introns are eventually spliced out (i.e. transcripts with retained introns cannot be exported from the nucleus). We observed a large range in the number of pre-mRNA observed per cell (from 0 to ~20), which also varied from intron to intron (*Figure 1—figure supplement 1B*).

As a corroboration of RNA FISH dispersal as a metric to quantify distal post-transcriptional splicing, we turned to nascent RNA sequencing, using a combination of metabolic labeling and cellular fractionation to capture RNA that are both newly synthesized and co-sediment with the chromatin fractions of cells (*Drexler et al., 2020*; *Figure 1—figure supplement 3*). We found a modest association between splicing index and both mean dispersal and transcription site size, suggesting that sequencing of nascent RNA was at least consistent with RNA FISH assessments of distal post-transcriptional splicing; however, more work is required to fully establish a quantitative correspondence between the metrics.

It is possible that pre-mRNA away from the transcription site are ultimately degraded without ever being spliced and thus represent an alternative 'dead-end' fate for the pre-mRNA rather than an intermediate on the path to a mature mRNA. This alternative possibility is difficult to eliminate with data relying on a snapshot in time, and hence studies using sequencing or fixed cell imaging (such as ours) suffer from this limitation. The further development of live-cell imaging capable of tracking individual mRNA throughout their lifetime may help resolve these issues. However, we do note that pre-mRNA contain fewer introns as they increase in distance from the transcription site (*Figure 1—figure supplement 2D*), which is at least consistent with our assumption that these introns are spliced as they travel away from the transcription site. Furthermore, pulse-chase experiments have shown that the 'yield' of splicing events is high for mRNA, suggesting that most pre-mRNA do not go down a dead-end pathway (*Eser et al., 2016*; *Wachutka et al., 2019*). Given this assumption, the dispersion we observe suggests that at least one intron of each observed gene is spliced partly post-transcriptionally.

## Different introns within the same gene are spliced largely independently of each other

In these experiments, we observed individual pre-mRNA with some but not all introns retained, and wondered whether these introns could be spliced in any order. Thus, we quantified the number of transcript intermediates that contained different introns in the same RNA molecule. For each RNA containing a pair of introns A and B, there are two alternative routes to generate a fully spliced mature mRNA: either intron A is spliced before intron B, or intron B is spliced before intron A. Were the splicing of these introns ordered, for instance if B always gets spliced after A, then we would expect the vast majority of partially spliced transcripts to contain intron B only, and that very few transcripts would contain intron A without intron B. We thus identified and counted partially spliced transcripts which contain only one intron without the other. To estimate the confidence in our measurements, we labeled the fourth intron of *TM4SF1* with two colors; in this case, any transcripts detected as containing one color without the other must be due to technical reasons, thereby setting a 'noise floor' (see Materials and methods for details). The dye used to label each intron also did not affect these measurements (*FKBP5* introns 8 and 9 dye-swap, rows 13 and 14 of *Supplementary file 1B*). It was also difficult for us to discriminate between models if both introns are spliced so rapidly that intermediates are not observed.

We observed a variety of patterns of splicing intermediates, some showing strong ordering (94% of intermediate transcripts of *FKBP5* contain intron 8 without intron 5) and some showing a lack of

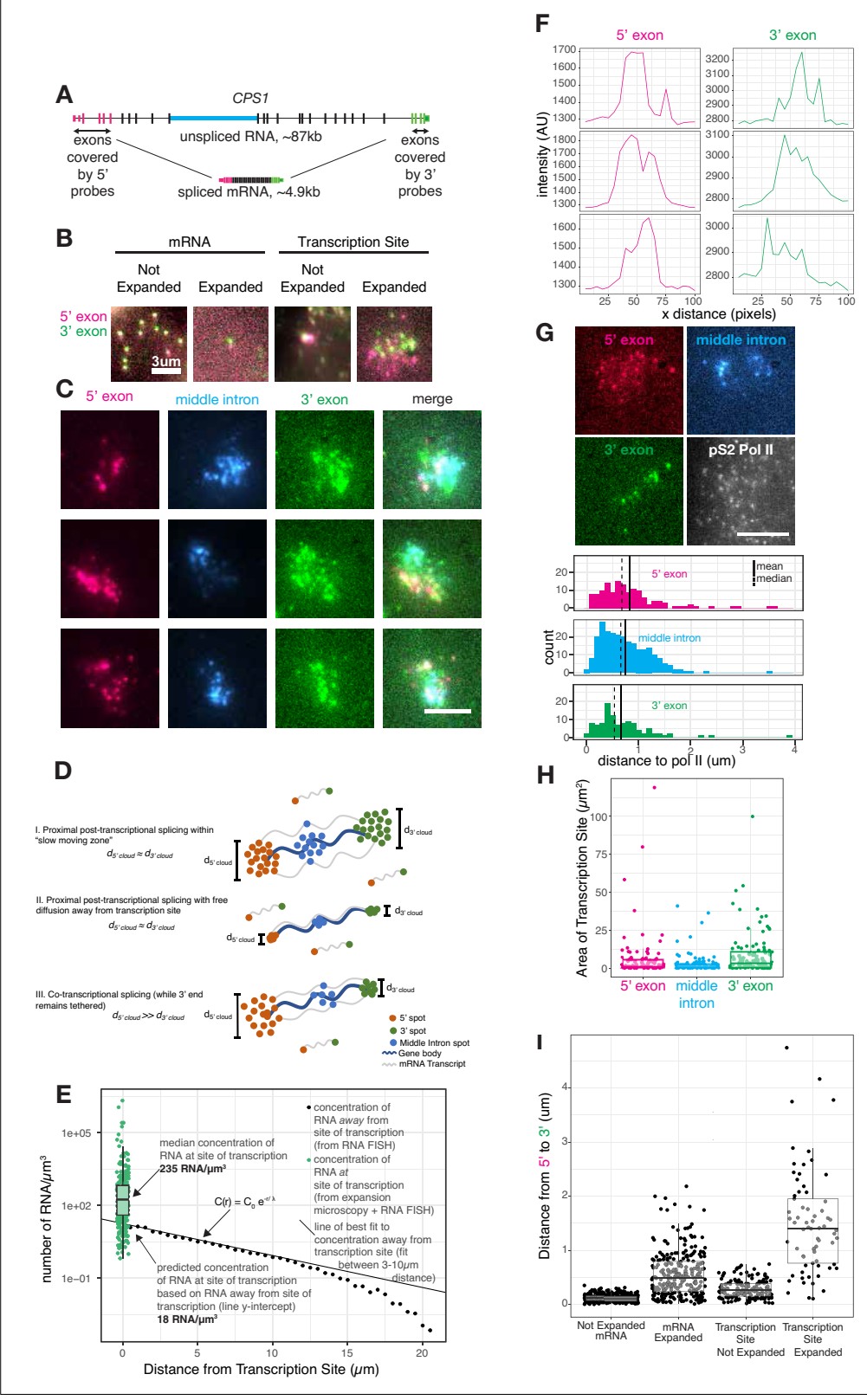

**Figure 2.** Transcripts dwell at the site of transcription after transcription is completed. (**A**) Gene and probe position diagram for CPS1 pre-mRNA and mRNA. (**B**) Example images of RNA FISH for CPS1 before and after expansion microscopy for individual mRNA and transcription sites (in HeLa cells, different representative cells shown in each image). (**C**) Example images of 5′, 3′, and middle intron CPS1 RNA FISH after expansion. (**D**) Schematic depicting

*Figure 2 continued on next page*

*Figure 2 continued*

simulated RNA FISH signal for three potential scenarios: a slow-moving zone around the site of transcription (I), free diffusion away from the transcription site (II), or co-transcriptional splicing of tethered transcripts (III). (**E**) Schematic depicting RNA FISH signal for diffusion or a slow-moving proximal zone for completed (but not necessarily spliced) transcripts. (**F**) Line scans of fluorescence intensity (arbitrary units) from transcription sites like in C. (**G**) Example images of co-IF (polymerase II) and RNA FISH for CPS1 after expansion. Quantification of images like that in G, and others, representing distance from each RNA spot to the nearest polymerase II IF spot. (**H**) Graph depicting area (in pixels) of polygons drawn around individual transcription sites after expansion. (**I**) Distance from 5' to 3' of expanded and unexpanded mRNA and transcription sites, as detected by CPS1 RNA FISH. This distance was calculated using nearest neighbors with replacement. All scale bars represent 5 µm unless otherwise noted. All expanded images and calculations are shown in expanded space scale (reduced by 4.65× to yield original space scale).

The online version of this article includes the following figure supplement(s) for figure 2:

**Figure supplement 1.** Expansion microscopy yields a 4.65-fold linear expansion and expands isotropically.

**Figure supplement 2.** Stochastic optical reconstruction microscopy (STORM) validation and post-3' RNA FISH.

**Figure supplement 3.** Intron characteristics do not influence dispersal.

---

ordering (50.3% of intermediate transcripts of *EEF2* contain intron 2 without intron 1). Lack of ordering between introns did not depend on distance between them, with no ordering observed for the neighboring *EEF2* introns 1 and 2 as well as the distant *CSP1* introns 1 and 34.

Overall, our data support a model in which splicing of different introns within the same primary transcript can occur in any order.

## Transcripts are untethered to the transcription site and move through a slow-moving transcription-site-proximal zone where splicing occurs after transcription is completed

While conventional single-molecule RNA FISH allowed us to determine what portion of splicing is happening post-transcriptionally and far from the site of transcription (*Supplementary file 1A*, *Figure 1—figure supplement 2B*), the resolution limits of conventional light microscopy made it impossible to distinguish whether transcripts are being spliced during the process of transcription (co-transcriptional) or after the completion of transcription but before the pre-mRNA moves away from the site of transcription (proximal post-transcriptional). Conventional light microscopy cannot easily distinguish these possibilities because all the RNA at or near the site of transcription are sufficiently close together that they are typically only visible as one large transcription site spot (see *Figure 1B* for example). It is therefore possible that lower mobility of transcripts near the transcription site may lead to the classification of proximal post-transcriptional splicing as co-transcriptional when using low-resolution imaging methods.

We thus used expansion microscopy to physically expand the transcription site (by around 4.6×, *Figure 2—figure supplement 1*) followed by staining by RNA FISH (*Figure 2A*) and imaging, thereby separating the single bright transcription site blob into visually distinct individual RNA intermediates at the site of transcription (*Figure 2B*; *Chen et al., 2016*; *Chen et al., 2015*). We labeled the 5' and 3' regions, as well as one interior intron, of the gene *CPS1*, for which the unspliced transcript would be quite long (~87 kb) but the spliced transcript is comparatively short (~5 kb) (*Figure 2A*). This labeling scheme allowed us to measure the locations of 5' exons, a middle intron, and 3' exons of pre-mRNA or processed mRNA in the vicinity of the expanded transcription site. Based on previous work (*Chen et al., 2016*), we expected expansion of transcription sites to be isotropic. To confirm isotropic expansion in our system, we imaged the same cell before and after expansion with and without a perturbation that changed the localization of introns (pladeinolide B treatment, which inhibits splicing and causes the introns of *EEF2* to form a well-defined blob, see Figure 4) and indeed observed that both the transcription site and the blob of *EEF2* introns expanded isotropically (*Figure 2—figure supplement 1B*, displaying the same cells before and after expansion). To further ensure that expansion microscopy did not alter the morphology of the transcription site, we confirmed that the transcription sites in expanded samples were the same size as those imaged using stochastic optical reconstruction microscopy (STORM) (*Rust et al., 2006*), a super-resolution imaging technique that does not rely on expansion (*Figure 2—figure supplement 2A and B*).

Lower mobility of transcripts near the site of transcription may lead to a buildup of multiple transcripts in the vicinity of the transcription site. Upon expanding, labeling, and imaging, we indeed observed that the 5', 3', and middle intron probe signals formed small clouds where we had previously observed the tight transcriptional focus (*Figure 2C*, quantified in *Figure 2—figure supplement 2C*). Of note, the expanded transcription site contains multiple 3' and 5' spots, while expanded cytoplasmic mRNA contain one 3' and 5' spot, indicating the presence of multiple transcripts at the transcription site as opposed to a single transcript broken into multiple parts as a result of expansion (*Figure 2B*). The variable brightness between spots in expanded versus unexpanded samples may have been due to variable lengths of transcribing RNA or RNA fragmentation during expansion, as each RNA is likely linked to the gel in multiple locations.

The presence of these clouds precluded any model of immediate, free movement of the transcript away from the RNA polymerase upon completion of transcription. If transcripts freely diffused away right after their transcription, then a mathematical model would predict an exponential decrease in pre-mRNA concentration with increasing distance away from the transcription site (*Maire and Youk, 2015*) (see Materials and methods for details, *Figure 2D* scenario II). We fit such a model to pre-mRNA that were between 3 and 10 μm away from the site of transcription, in which range the diffusion model fits well. From these data, we were able to estimate what the concentration should be at or near the transcription site given the free diffusion model (essentially, the y-intercept of the concentration curve). We found that the predicted concentration (18 molecules per cubic micron) was far lower than the actual concentration (235 molecules per cubic micron) in the vicinity of the transcription site measured by expansion microscopy (*Figure 2E*). This high concentration in the 'cloud' is not consistent with free diffusion subsequent to the completion of transcription (*Figure 2D* scenario II), but is compatible with the existence of a distinct 'proximal zone' surrounding the transcription site through which pre-mRNA move more slowly than in the more distant nucleoplasm (*Figure 2D* scenario I).

We wondered whether the pre-mRNA present in this zone were partially spliced intermediates or full pre-mRNA, the former signifying co-transcriptional splicing and the latter signifying predominantly post-transcriptional splicing. If splicing occurred concurrently or shortly after transcription, then we would expect to see a potential cloud of 5' ends but a tight spot of 3' ends corresponding to the point of transcriptional termination (*Figure 2D* scenario III). However, if splicing occurred some time after the entire full-length pre-mRNA was transcribed, one would expect to see separate clouds for both the 5' and 3' ends, representing pre-mRNA that have completed transcription and are slowly diffusing through the proximal zone (*Figure 2D* scenario I). We observed similarly sized clouds for both the 5' and 3' ends of the pre-mRNA in the proximal zone. Since RNA polymerase II can transcribe beyond the poly-A site (*Anamika et al., 2012*; *Core et al., 2008*), it may be possible that observed clouds are a result of transcripts tethered beyond the 3' poly-A site of the gene body. To eliminate this possibility, we performed RNA FISH for a sequence 2 kb beyond the 3' poly-A site of *CSP1*. We found that the 3' and post-3' transcription sites were also approximately the same size, suggesting these transcripts are not tethered to chromatin as splicing is occurring (*Figure 2—figure supplement 2D and F*).

Based on the size of these clouds, we estimated the diameter of this cylindrical slow-moving zone around the gene body to be around 0.3 μm (*Figure 2F*, see Materials and methods for calculations). This width does not depend on the length of an individual transcript or gene body, but rather extends radially outward from the gene body.

We concluded that pre-mRNA are not spliced immediately upon transcription, but rather that a large proportion of the splicing occurs post-transcriptionally while the pre-mRNA were moving through the proximal zone. (Splicing that occurred while in this zone would appear to be co-transcriptional by conventional single-molecule RNA FISH.) To further test for the occurrence of splicing while pre-mRNA were in this zone, we simultaneously labeled an intron while labeling the 5' and 3' ends of the pre-mRNA. Similar to the signals from the 5' and 3' ends, the intronic signal also formed a cloud, showing that splicing has not yet been completed as the pre-mRNA move through this proximal zone (*Figure 2G and H*). Furthermore, the clouds of the 5' and 3' ends of the pre-mRNA are typically non-overlapping and are further apart than the mature mRNA we found in the cytoplasm, suggesting that these clouds do not represent fully mature, spliced mRNA (*Figure 2I*).

We next wanted to confirm the presence of this slow-moving zone in a live cell as transcription is actively taking place, as this method does not require manipulation through fixation or expansion. We obtained data from cells in which the 3' UTR of the gene *TFF1* was labeled with MS2-GFP (*Wan et al.,*

*2021*), allowing us to observe the geometry of transcription sites in living cells (*Figure 3—videos 1–3*). If transcripts immediately and rapidly diffused away from the transcription site after synthesis at the same diffusion rate as in the nucleoplasm, then the 3' end of the transcripts would form a tight, diffraction limited spot at the site of transcription, much like the 3' ends of individual transcripts we detected by mRNA FISH (*Figure 2B*). (Note that the limit of detection of MS2-GFP was such that only transcription sites, presumably with multiple transcripts on them, would be detectable; hence, individual mRNA were not detectable.) If, on the other hand, there were a gene-proximal zone in which diffusion was slower, one would expect to see primary transcripts at a higher concentration within the zone (see *Figure 2E*). This higher concentration would translate into a cloud of primary transcripts visible as a blob that was larger than the diffraction limit, reflecting the size of the putative slow-moving zone. Thus, the presence of transcription site spots larger than the diffraction limit would be an indicator of the presence of a cloud of primary transcripts and hence a slow-moving zone around the site of transcription.

It is challenging to rigorously discriminate between a diffraction limited spot and a blob with larger spatial extent based solely on the area of the fluorescent region because the thresholds chosen in determining area can be somewhat arbitrary, and the underlying data is often noisy. Hence, we instead used eccentricity (a measure of deviation from a perfect circle, ranging from 0 to 1) as a means to distinguish diffraction limited spots from blobs, reasoning that blobs larger than the diffraction limit could potentially be asymmetric, thus exhibiting increased eccentricity as compared to pure diffraction limited spots. Visually, several images showed evidence for such asymmetry (*Figure 3A–C*). We quantified the asymmetry by measuring the eccentricity for 205 transcription site images from 36 total temporal transcription site tracks (*Figure 3D*). We found that the majority of images (141 of 205) showed evidence of substantial eccentricity (defined as eccentricity >0.6) (*Figure 3C*). As these sites are unambiguously larger than a diffraction limited spot, we concluded that RNA were likely retained at the site for some time following transcription, corroborating the existence of the slow-moving zone.

It should be noted that not all transcription site images appeared eccentric. Possible reasons are that the transcription burst duration was short enough that there was not time for a buildup of pre-mRNA in the slow-moving zone, or that the blob happened to grow in a symmetric fashion, or that the three-dimensional (3D) orientation of the asymmetry was such that it was not detectable in the single x-y plane of the microscope.

Similarly, in our expansion microscopy data, the relative positions of the 5', 3', and middle intron clouds adopted a wide variety of conformations, suggesting that the linear genomic order of the pre-mRNA may not be strictly maintained during transit through the proximal zone (*Figure 2C*). The length of each intron, as well as the strength of its 3' and 5' splice site, did not affect its dispersal from the transcription site, suggesting that transcript size and splice site sequence are not major contributors to the dynamics of the slow-moving zone (*Figure 2—figure supplement 3*). Also, neither the 5' exon nor 3' exon signals overlapped with actively elongating RNA polymerase II immunofluorescence signals (see Materials and methods), potentially suggesting the relatively few mRNA spots that are quite close to RNA polymerase II signal are undergoing active transcription, while the majority of pre-mRNA in the slow-moving proximal zone are not being actively transcribed (*Figure 2G*). It is important to note, however, that we do not label the entirety of the pre-mRNA molecule, nor can we be sure that we can detect single molecules of RNA polymerase II, thus it is difficult to eliminate the possibility that some other part of the pre-mRNA is close to RNA polymerase II or that a pre-mRNA is localized to an undetected single RNA polymerase II molecule.

Another interesting observation was that the size of 5', intron, and 3' clouds, the number of spots in each cloud, and the area of live-cell transcription sites varied considerably from cell to cell (*Figures 2C, H and 3A, B*). In some cases, we observed more 5' spots than 3' spots, which may be due to the interception of an ongoing transcriptional burst (where RNA polymerase II has already synthesized the 5' ends of the RNA but has not yet synthesized the 3' ends). We also observed some cases where there were more 3' spots than 5' spots and indeed many cases where the 3' clouds are more dispersed than the 5' clouds, which may suggest that the spots of 5' probe may represent a pileup of transcripts at the 5' end, such that we only see one bright spot where we should see several. When the dyes of the two probes are swapped, the 3' clouds are still consistently larger than the 5' clouds, suggesting that it's unlikely to be a dye-bias effect (*Figure 2—figure supplement 1C*).

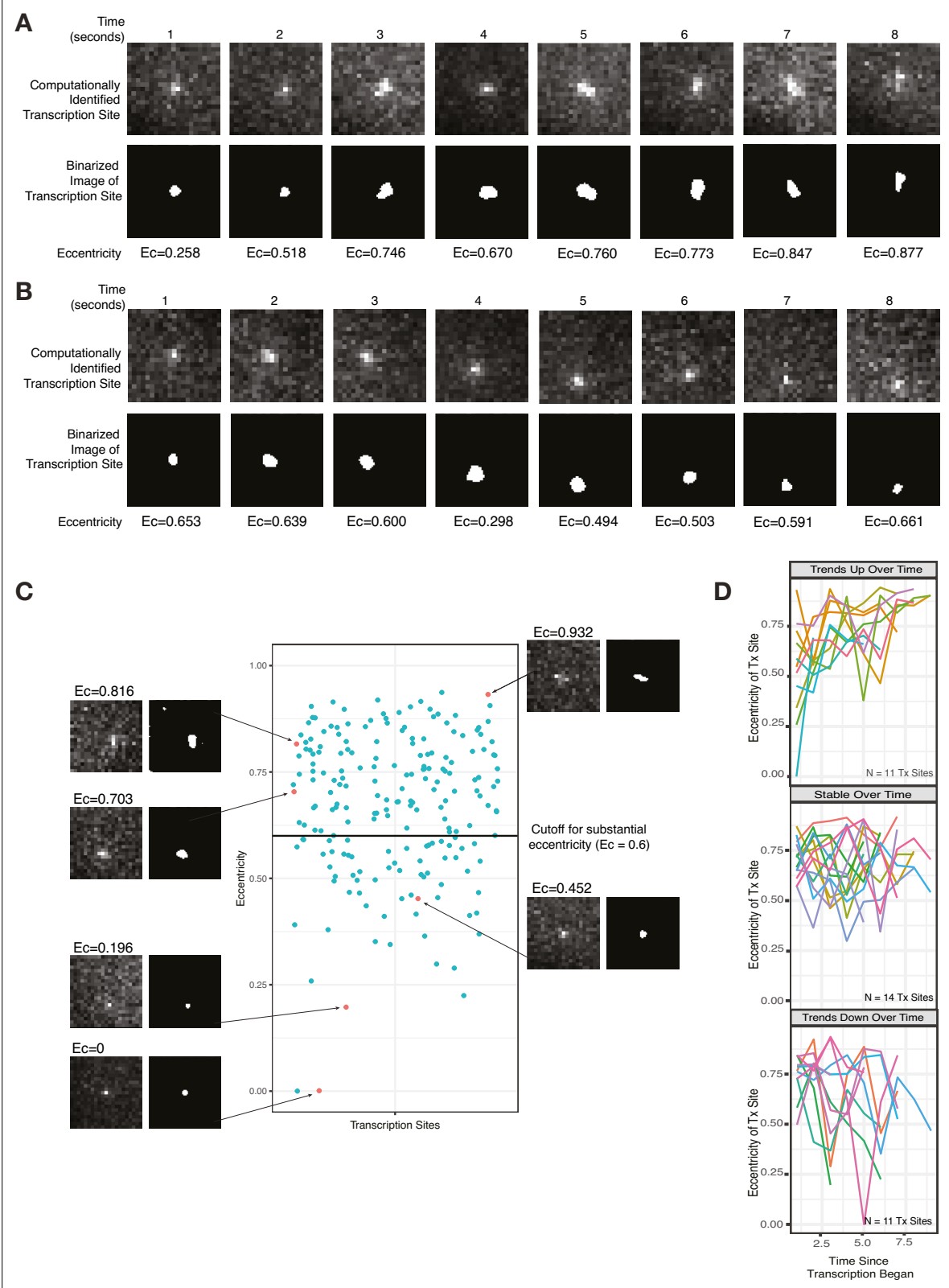

**Figure 3.** Live-cell imaging corroborates a slow-moving zone around the site of transcription. (**A–B**) Representative images from two cells of a transcriptional burst of TFF1 labeled with MS2-GFP. Each image is taken 1 s apart. Shown are the raw, computationally identified transcription site images, as well as the binarized image used to calculate eccentricity. (**C**) Plot of eccentricity for each transcription site image, alongside representative images that show both eccentric and non-eccentric sites. (**D**) Graphs of eccentricity measured in individual transcription sites tracked over time.

*Figure 3 continued on next page*

*Figure 3 continued*

The online version of this article includes the following video(s) for figure 3:

**Figure 3—video 1.** Representative example of live-cell videos analyzed in *Figure 3*.

https://elifesciences.org/articles/91357/figures#fig3video1

**Figure 3—video 2.** Representative example of live-cell videos analyzed in *Figure 3*.

https://elifesciences.org/articles/91357/figures#fig3video2

**Figure 3—video 3.** Representative example of live-cell videos analyzed in *Figure 3*.

https://elifesciences.org/articles/91357/figures#fig3video3

Thus, expansion microscopy revealed that after the completion of transcription, pre-mRNA move slowly through a slow-moving proximal zone, during which splicing may be ongoing. This finding potentially unites existing conflicting reports of co-transcriptional versus post-transcriptional splicing, as splicing within the slow-moving zone would appear co-transcriptional when evaluated by conventional light microscopy but in reality occurs after transcription has been completed.

## Localization of unspliced pre-mRNA to speckle-proximal compartments is gene specific

We wondered where transcripts went after they were released from the slow-moving transcription-site-proximal zone that was revealed by expansion microscopy (see *Figure 2C*). We hypothesized that the transcripts could do one of three things (*Figure 4A*):

1. freely diffuse away from the transcription-site-proximal zone through the nucleoplasm (nuclear dispersal);
2. be tethered to the transcription-site-proximal zone in some manner (tethering);
3. fill a compartment, potentially around or adjacent to the transcription-site-proximal zone or other nuclear bodies (compartmentalization).

Owing to the lack of dispersal of some introns, relatively few unspliced RNA were detectable outside of the transcription-proximal zone, making it difficult to discriminate between these hypotheses. Thus, we inhibited splicing to generate more pre-mRNA, making it easier to track their localization after leaving the transcription-site-proximal zone. Upon splicing inhibition, we observed three distinct trafficking behaviors for pre-mRNA species: one in which there are increased numbers of dispersed pre-mRNA throughout the nucleus (nuclear dispersal; consistent with scenario 1), one in which the pre-mRNA are located in a large blob, likely around the transcription-site-proximal zone or another nuclear body (blobs; scenario 2 or 3), and one in which the pre-mRNA dispersal pattern looked identical with or without splicing inhibition (non-splicing inhibited) (*Figure 4B and C*).

The compartmentalization pattern only appeared for 3 out of the 16 genes we tested (*EEF2*, *GAPDH*, and *RPL13A*), whereas we observed a nuclear dispersal phenotype for 7 genes and no change in dispersal for 6 genes. To test for potential tethering of transcripts to some location in the nucleus, we labeled the 5′ and 3′ ends of the pre-mRNA in different colors, reasoning that if either end of the pre-mRNA was tethered to a particular location, then the signal from that particular end would form a tighter spot in the nucleus while the other end would fill the compartment (*Figure 4D*, *Figure 4—figure supplement 1A*). We found, however, that the 5′ and 3′ ends of the pre-mRNA both filled the entire blob, suggesting that the pre-mRNA spread to fill the entire putative compartment (as in scenario 3 described above, compartmentalization) (*Figure 4A and D*).

We wondered if these compartmentalized pre-mRNA were located near nuclear speckles, which are compartments in the nucleus that contain concentrated splicing and transcription factors (*Zhang et al., 1994*). To test this hypothesis, we performed RNA FISH simultaneously with immunofluorescence for SC35, a component of speckles, and saw that these compartmentalized pre-mRNA did indeed appear near nuclear speckles both before (*Figure 4—figure supplement 1C*) and after (*Figure 4G*) splicing inhibition.

We corroborated the spatial association we observed between speckles and compartmentalized pre-mRNA by analyzing previously published high-throughput sequencing data (Tyramide Signal Amplification [TSA]-Seq 2.0) that quantified the distance of genes from speckles and other nuclear compartments (*Chen et al., 2018*; *Zhang et al., 2019*). We found that even in the absence

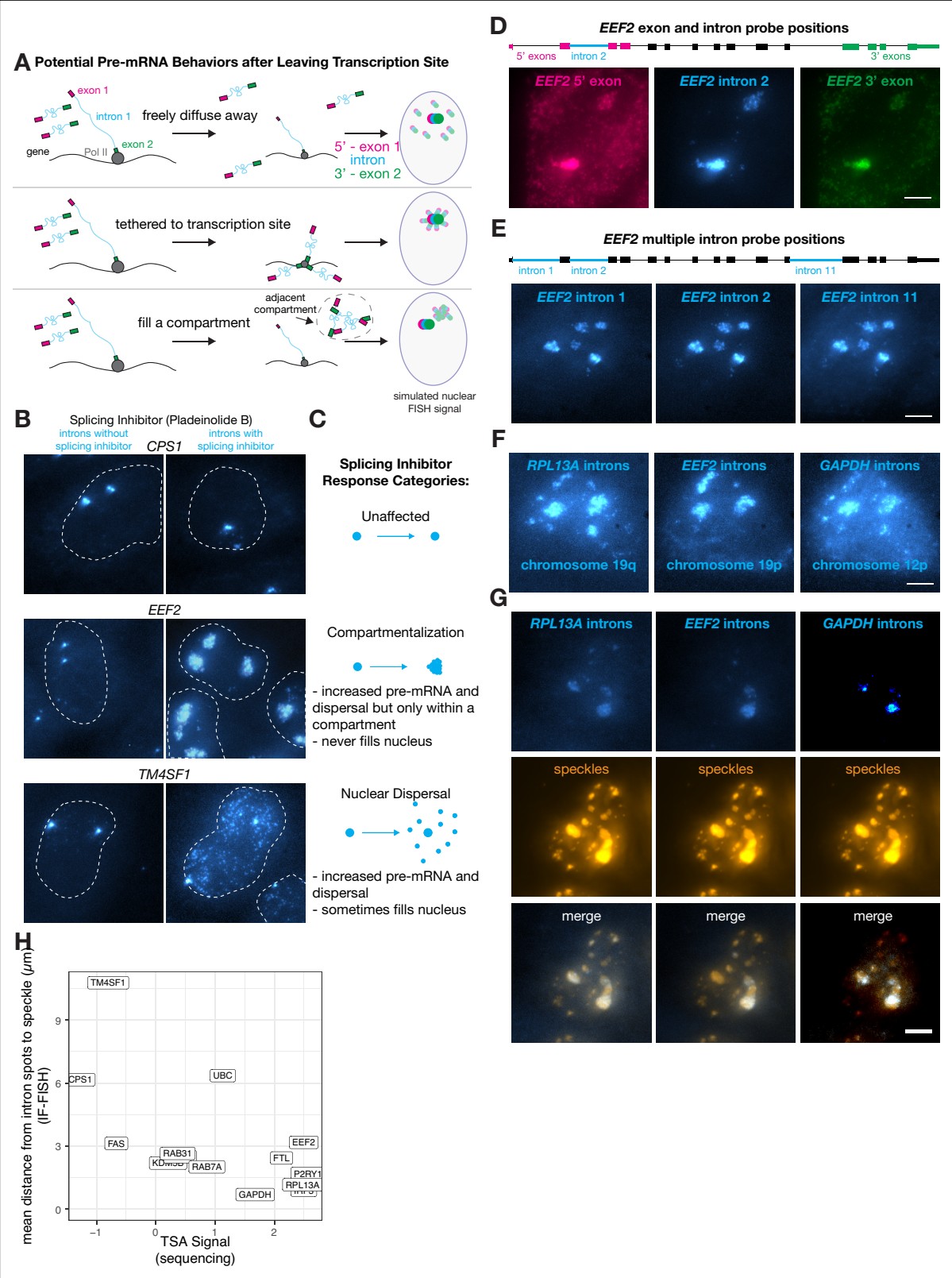

**Figure 4.** Some transcripts localize to a speckle-proximal compartment after splicing inhibition. (**A**) Schematic of possibilities for RNA movement post- transcription. (**B**) Example images of introns of specified genes before and after pladeinolide B treatment (treatment status as indicated in figure). Outline represents nucleus. (**C**) Description of three response types to pladeinolide B. (**D**) EEF2 RNA FISH exon and intron images with pladeinolide B treatment. Scale bar = 5 μm. (**E**) EEF2 RNA FISH intron only images with pladeinolide B treatment. Scale bar = 5 μm. (**F**) RPL13A, GAPDH, and EEF2

*Figure 4 continued*

RNA FISH intron images with pladeinolide B treatment. Scale bar = 5 μm. (**G**) Co-staining of RNA FISH for specified intron and SC35 IF in the same cell treated with pladeinolide B. Scale bar = 5 μm. (**H**) Quantification of speckle decile from previously published data (Tyramide Signal Amplification [TSA]-Seq 2.0), compared with distance from nearest speckle calculated based on RNA FISH of specified genes without pladeinolide B treatment.

The online version of this article includes the following figure supplement(s) for figure 4:

**Figure supplement 1.** Compartmentalization genes before splicing inhibition.

of pladeinolide B, the genes that localized to compartments post-splicing inhibitor treatment were indeed the closest to speckles. Furthermore, the distance from speckles (without splicing inhibitor treatment) was anticorrelated with the signal from TSA-Seq 2.0, which measures the distance from all genes to various physical anchors in the nucleus (in this case, speckles). Those data showed that three compartmentalized genes were all within the most speckle-associated transcripts, while all other tested genes (both nuclear dispersal and non-responsive; 11 genes) exhibited a much broader range of distances to speckles (*Figure 4H*). This anticorrelation is to be expected, because the genes closest to speckles will receive the most reads in TSA-Seq and should have the smallest distance to speckles as measured by IF-FISH.

Speckles form a set of subcompartments within the nucleus. We thus wondered whether pre-mRNA from the genes exhibiting 'compartmentalization' in their post-transcriptional trafficking would go to all of these speckle compartments, or rather just a gene-specific subset. To test these possibilities, we performed RNA FISH on multiple introns within the same 'compartmentalization' gene (*EEF2*) as well as introns from several different 'compartmentalization' genes simultaneously (*EEF2*, *GAPDH*, and *RPL13A*). We observed that multiple introns retained in pre-mRNA from the same gene colocalized to the same subset of speckles (*Figure 4E*), suggesting that all unspliced pre-mRNA from a particular gene localize to the same subset of speckles. We also observed that pre-mRNA from multiple 'compartmentalization' genes (*EEF2*, *GAPDH*, and *RPL13A*) localize to a similar set of speckles after splicing inhibition (*Figure 4F*), although there are some differences. However, when observing the intron distributions of pre-mRNA from *EEF2*, *GAPDH*, and *RPL13A* before splicing inhibition, they do not appear to colocalize with one another or with the same speckles (*Figure 4—figure supplement 1C*), suggesting perhaps that splicing inhibition impacts these responses. Overall, our results suggest that mRNA trafficking upon leaving the slow-moving zone is gene specific.

## Dispersal is not an inherent trait of individual introns and can vary with transcription level

We wondered whether the degree of dispersal (and thus, the degree of post-transcriptional splicing) was an inherent property of each intron or whether dispersal could vary due to other factors such as the level of transcription. To test whether the level of transcription affected the degree of dispersal, we treated A549 cells with dexamethasone to induce transcription of the gene *FKBP5*, and then performed RNA FISH against introns 1, 8, and 9 at various time points in dexamethasone to measure the degree of dispersion (*Figure 5A and B*). We saw an increase in both exon and intron spot counts (*Figure 5B*) over time and a corresponding increase in the dispersal of some, but not all, introns (*Figure 5C*).

The fact that intron dispersal increased with transcription level for at least some introns shows that dispersal is not an inherent property of each intron but can depend on other variables like the level of transcription (*Figure 5C*). Intron 9 did not exhibit an increase in dispersal even with long exposure to dexamethasone (8 hr) (*Figure 5C*).

We believe that the increased dispersal represents an escape of unprocessed pre-mRNA from the site of transcription. This increase would only happen in cases where splicing was slow enough that splicing of all introns was not completed before termination of transcription, therefore allowing time for these pre-mRNA to disperse away from the site of transcription. The lack of dispersal of some introns, even with increased transcription, suggests that these introns are spliced so quickly that pre-mRNA containing those introns have no time to disperse away from the site of transcription even in case of increased transcription.

We hypothesized that the increased dispersal of some introns during periods of increased transcription may be due to a local depletion of splicing factors. A local depletion of splicing factors

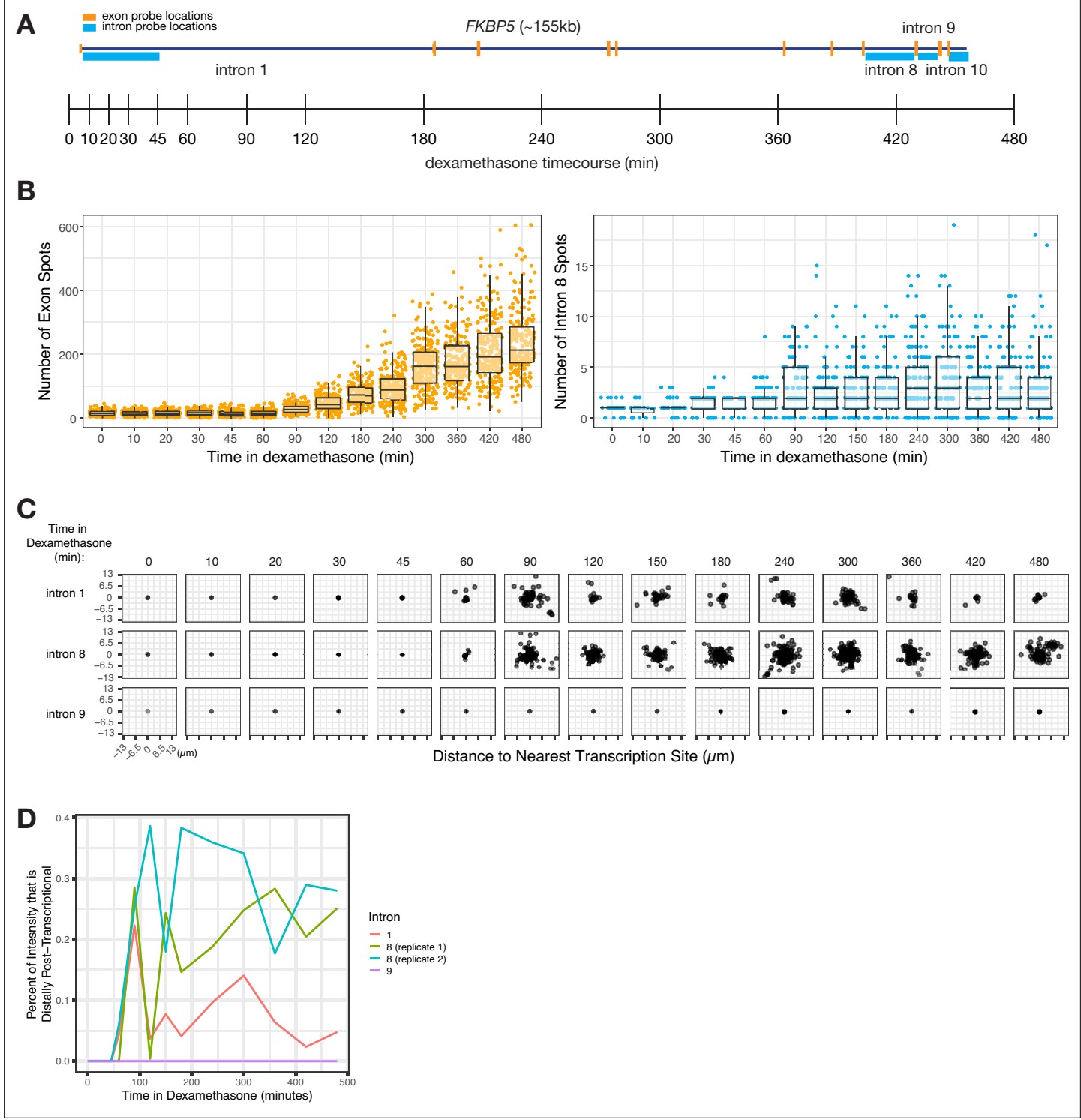

**Figure 5.** Intron dispersal varies with transcription level and is therefore not an inherent property of each intron. (**A**) Gene and probe diagram for FKBP5 and schematic of dexamethasone treatment schedule. (**B**) Quantification of FKBP5 RNA FISH exon and intron spots over time of treatment in dexamethasone (in A549 cells). (**C**) Dispersal graphs (as quantified from RNA FISH) of FKBP5 introns 1, 8, and 9 over time in dexamethasone (in A549 cells). (**D**) Graph showing the percentage of intron intensity that is distally post-transcriptional for three FKBP5 introns over time in dexamethasone.

could occur as more pre-mRNA fill the local area and absorb the local pool of splicing factors (or particular splicing cofactors), resulting in more pre-mRNA escaping from the transcription-proximal region before undergoing splicing. If the concentration of splicing factors remained constant (i.e. no local depletion), we would expect the rate of splicing to remain constant, and the percentage of unspliced pre-mRNA far from the transcription site to be unchanged, regardless of transcription rate. Instead, we observed a significant increase in the proportion of intron signal for two introns over time (from 0% to 22% for intron 1 and from 0% to 29% for intron 8; *Figure 5D*), suggesting a possible local depletion of splicing factors at the transcription site. (Of note, we may potentially miss post-transcriptional splicing occurring after a burst has concluded.) These increases in the percent of distal post-transcriptional splicing show that even the same intron can exhibit a range of spatiotemporal patterns of splicing depending on expression levels and potentially on limiting splicing factors in the vicinity of the gene itself.

## Discussion

By using expansion microscopy combined with single-molecule RNA FISH, we were able to obtain direct measurements of the spatial distribution of splicing intermediates in the vicinity of the transcription site. We found evidence for extensive post-transcriptional splicing that occurs throughout that zone. Our results support a model that unifies many of the observations made of both co-transcriptional and post-transcriptional splicing. We argue that splicing can occur continuously and post-transcriptionally while pre-mRNA move through this slow-moving zone. Fractionation approaches may have interpreted such splicing intermediates as arising co-transcriptionally owing to proximity to the transcription site. We also found introns that seemed to be spliced out so close to the site of transcription that we were unable to distinguish whether they were spliced co- or post-transcriptionally. As methods are developed with ever-increasing spatiotemporal resolution, it may soon be possible to discriminate splicing dynamics for these introns as well.

Our results suggest that pre-mRNA are not tethered to the site of transcription while they move through this transcription-site-proximal compartment, in contrast to what is suggested by Dye et al., where the authors suggest that exons are tethered to polymerase II as splicing is occurring (*Dye et al., 2006*). The lack of colocalization of introns with polymerase II also suggests that splicing is not happening close to polymerase II. This conclusion stands in contrast to the data of *Alexander et al., 2010* which suggested that almost all splicing occurs while polymerase II is still paused proximal to the intron that was recently transcribed (*Vargas et al., 2005*). This discrepancy may be due to species-specific differences. In yeast, polymerase II has been found to pause at several sites involved in splicing, including the terminal exon, the 3′ SS, and internal exons (for a review, see *Carrillo Oester-reich et al., 2011*). It is also possible that increased resolution made possible by expansion microscopy has allowed us to detect previously indistinguishable distances between introns and polymerase II.

Our model stands in contrast to some number of prior studies of the timing of splicing, which suggest that splicing happens immediately (within 15–20 s or 45–100 nucleotides of transcription) after transcription of each intron is completed (*Oesterreich et al., 2016*; *Reimer et al., 2021*; *Martin et al., 2013*; *Eser et al., 2016*; *Wallace and Beggs, 2017*; *Alpert et al., 2017*; *Huranová et al., 2010*; *Baurén and Wieslander, 1994*). This discrepancy may arise from a variety of reasons, including but not limited to species-specific differences and the use of different assays to measure the timing of splicing (a thorough review is available in *Alpert et al., 2017*). Of note, we see that increased transcription level correlates with intron dispersal, suggesting that the percentage of splicing occurring away from the transcription site is regulated by transcription level for at least some introns. This may explain why we observe post-transcriptional splicing of all genes we measured, as all were highly expressed. Our model is in agreement with fractionation work by Pandaya-Jones et al. and by Drexler et al. that suggest that specific transcripts are retained in the chromatin and that splicing is not completed until transcription has finished (*Drexler et al., 2020*; *Pandya-Jones et al., 2013*). Our model is also in agreement with several other studies which suggest that there can be significant delays between RNA transcription and splicing (*Wachutka et al., 2019*; *Honkela et al., 2015*; *Hao and Baltimore, 2013*; *Singh and Padgett, 2009*; *Rabani et al., 2014*; *Windhager et al., 2012*; *Darnell, 2013*).

What might the slow-moving zone represent, physically? A natural candidate is the chromatin in the vicinity of the gene itself. Live imaging studies have shown that mRNA exhibit slow diffusion in chromatin-dense regions (*Vargas et al., 2005*). Furthermore, recent work tracking cells over time

by sequencing shows that mRNA spends relatively more time on chromatin than in the nucleoplasm (*Smalec et al., 2022*), further lending credence to this hypothesis.

Our expansion microscopy results suggest that the distance between the 5' end and the 3' end of RNA at the site of transcription is greater than the distance between the 5' end and 3' end of mRNA. This increased 5' to 3' distance suggests that the transcripts at the site of transcription are either unpackaged (perhaps due to decreased RNA binding proteins occupancy) or are simply longer because there are likely more introns incorporated into the transcripts close to the site of transcription than away from it due to progressive splicing over time. If this 5' to 3' distance increase were due to increased retention of introns, that would also support the post-transcriptional splicing model. We note that it is difficult to know which particular 5' and 3' spots correspond to the same pre-mRNA molecule near the site of transcription; further expansion or live tracking of individual mRNA might help make those connections and refine our measurement of distances between 5' and 3' ends of transcripts.

One of the original models for splicing is the 'first come, first serve' model in which each intron is immediately spliced upon the completion of transcription, in a 5' to 3' order (*Aebi and Weissman, 1987*). Our results suggest that first come, first serve is not the case, based on seeing low splicing rates (or high dispersion) for even the 5' most introns of some genes. This lack of first come, first serve splicing is confirmed by others in several different situations (*de la Mata et al., 2010*; *Yang et al., 2012*; *Kessler et al., 1993*). Our results also suggest that introns are spliced independently of one another, whether they are genomically proximal or distal to each other. This conclusion is in contrast to other work which suggests splicing of particular introns is controlled or gated by the splicing of other introns or exons within the same gene (*Drexler et al., 2020*; *Choquet et al., 2022*; *Kim et al., 2017*).

Our splicing inhibition results show that the trafficking of transcripts after they escape the slow-moving transcription-proximal zone varies by gene. Our results were consistent with sequencing-based metrics, and those metrics are in turn largely consistent across multiple cell types, suggesting that the speckle-associative property is not subject to cell-type-specific regulation (*Chen et al., 2018*). The association between specific genes and speckles has been observed before in several studies by Jeanne Lawrence (*Shopland et al., 2002*; *Johnson et al., 2000*; *Smith et al., 1999*). Our observations are also consistent with those of Girard et al. in which certain genes are retained post-transcriptionally at speckles as splicing occurs and then are released and immediately exported from the nucleus (*Girard et al., 2012*). Wang et al. also observed speckle localization of RNA upon microinjection into nuclei (*Wang et al., 1991*). Our data further support the conclusion that speckle association can be gene specific and can help retain pre-mRNA in the gene's vicinity until post-transcriptional splicing is completed.

## Study limitations

With the notable exception of our live-cell data in *Figure 3*, the experiments in this study utilize fixed cells and capture transcription at a single moment in time. As a result, many of the conclusions we draw can be strengthened by future work that evaluates active transcription through live-cell imaging or other methods that do not involve fixation. Our investigation of intron localization in *Figure 4* relies upon the inhibition of splicing, which may impact several aspects of cellular physiology. This paper also does not address potential relationships between splicing and other co-transcriptional events, such as 5' capping or 3' end formation. Future work may unravel dependencies between these events and the splicing of introns, and a potential role for the transcription-site-proximal slow-moving zone in other aspects of mRNA processing. Lastly, a major outstanding question is what sequence-based features determine the distinct behaviors of different introns. Large-scale imaging and synthetic libraries of introns may be required to make such conclusions.

## Materials and methods

**Key resources table**

| Reagent type (species) or resource | Designation | Source or reference | Identifiers | Additional information |
|---|---|---|---|---|
| Antibody | Anti-SC35 (mouse monoclonal) | Abcam | ab11826 | IF: (1:200) |

*Continued on next page*

*Continued*

| Reagent type (species) or resource | Designation | Source or reference | Identifiers | Additional information |
|---|---|---|---|---|
| Antibody | Phospho S2 polymerase II (rat monoclonal) | Active Motif | 61083 | IF: (1:200) |
| Cell line (*Homo sapiens*) | HeLa | Lab of Dr. Phillip Sharp (MIT) | | |
| Cell line (*Homo sapiens*) | HeLa-S3 | ATCC | CCL-2.2 | |
| Cell line (*Homo sapiens*) | A549 | ATCC | CCL-185 | |
| Cell line (*Homo sapiens*) | CRL-2097 | ATCC | CCD-1079Sk | |
| Chemical compound, drug | Pladienolide B | Tocris Biosciences | 6070500U | |
| Chemical compound, drug | Dexamethasone | Sigma | D2915 | |
| Chemical compound, drug | 4sU (4-thiouridine) | Sigma | T4509 | |
| Sequence-based reagent | Single-molecule RNA FISH probes targeting CSP1, FKBP5, TM4SF1, and EEF2 | IDT DNA | | See *Supplementary file 2* for sequences |
| Software, algorithm | rajlabimagetools | GitHub; *Raj et al., 2018* | https://github.com/arjunrajlaboratory/rajlabimagetools | |
| Software, algorithm | KNIME | GitHub; *Larson and Wan, 2019* | https://github.com/CBIIT/Larson-Lab-CCR-NCI/tree/master/Wan_GeneTrap_2019/KNIME_Workflows | |
| Software, algorithm | STAR (v2.5.1a) | *Dobin et al., 2013* | | |

## Resource availability

### Lead contact
Further information and requests should be directed to and will be fulfilled by the lead contact, Arjun Raj (arjunrajlab@gmail.com).

### Materials availability
This study did not generate new materials.

## Method details

### Cell culture, splicing inhibition, and FKBP5 induction
HeLa (kind gift of the lab of Dr. Phillip Sharp, MIT) and A549 (human lung carcinoma, A549, ATCC CCL-185) cells were cultured in DMEM (Gibco) supplemented with 50 U/mL penicillin, 50 µg/mL streptomycin, and 10% fetal bovine serum (FBS, Fisher). Splicing inhibition was accomplished by treating HeLa cells with 1 µM pladienolide B (Tocris Biosciences, 6070500U) for 4 hr, as described by *Pandya-Jones and Black, 2009*. HeLa cells were then fixed and used for RNA FISH as described below. FKBP5 was induced by treating A549 cells with 25 nM dexamethasone (Sigma, D2915) for the specified lengths of time. A549 cells were then fixed and used for RNA FISH as described below.

### RNA fluorescence in situ hybridization, STORM, and expansion microscopy
Single-molecule RNA FISH was performed on samples as described previously (*Johnson et al., 2000*). Cells were fixed in 4% formaldehyde and permeabilized with 70% ethanol before in situ hybridization was performed using the probes described in *Supplementary file 2*. Samples were simultaneously co-stained with probes for the exon of gene of interest (labeled in cy3), two introns of the gene of interest (labeled in alexa594 or atto647N), and cyclin mRNA (labeled in either atto700 or atto647N)

(Stellaris oligonucleotides, Biosearch Technologies). Samples were then washed twice with 2× saline sodium citrate buffer (SSC) containing 10% formamide (Ambion), and then 2× SSC supplemented with DAPI (Molecular Probes D3571) to stain the cell nuclei. Cells were submerged in 2× SSC with DAPI for imaging. Chromatic aberration was evaluated using multi-color fluorescent beads and was far smaller than any biologically relevant distances we measured (see *Figure 1—figure supplement 2C*).

STORM was performed on the 5' exons and middle intron of CSP1 in HeLa cells. Cells were imaged in Vutara d-STORM imaging buffer (20 mM cysteamine [MEA]+1% 2-mercaptoethanol+1×Gloxy [glucose oxidase+catalase dissolved in 50 mM Tris-HCl+10 mM NaCl] in buffer B [50 mM Tris-HCl+10 mM HCl+10% glucose]). Samples were imaged on a Vutara VXL microscope and exported points were reconstructed in MATLAB. Transcription sites were identified by alignment with widefield images, and area was calculated using the polyarea() function in R.

For combined expansion microscopy and RNA FISH, expansion microscopy was performed as described by Chen et al. (*Nojima et al., 2015*). Briefly, Acryloyl-X, SE (6-((acryloyl)amino)hexanoic acid, succinimidyl ester, here abbreviated AcX; Thermo Fisher) was resuspended in anhydrous DMSO at a concentration of 10 mg/mL, aliquoted and stored frozen in a desiccated environment. Label-IT Amine Modifying Reagent (Mirus Bio, LLC) was resuspended in the provided Mirus Reconstitution Solution at 1 mg/mL and stored frozen in a desiccated environment. To prepare LabelX, 10 µL of AcX (10 mg/mL) was reacted with 100 µL of Label-IT Amine Modifying Reagent (1 mg/mL) overnight at room temperature with shaking. LabelX was subsequently stored frozen (−20°C) in a desiccated environment until use.

Fixed cells were washed twice with 1× PBS and incubated with LabelX diluted to 0.002 mg/mL in MOPS buffer (20 mM MOPS pH 7.7) at 37°C for 6 hr followed by two washes with 1× PBS.

Monomer solution (1× PBS, 2 M NaCl, 8.625% [wt/wt] sodium acrylate, 2.5% [wt/wt] acrylamide, 0.15% [wt/wt] *N,N*'-methylenebisacrylamide) was mixed, frozen in aliquots, and thawed before use. Prior to embedding, monomer solution was cooled to 4°C to prevent premature gelation. Concentrated stocks (10% wt/wt) of ammonium persulfate initiator and tetramethylethylenediamine accelerator were added to the monomer solution up to 0.2% (wt/wt) each. 100 µL of gel solution specimens were added to each well of a Lab Tek 8 chambered coverslip and transferred to a humidified 37°C incubator for 2 hr.

Proteinase K (New England Biolabs) was diluted 1:100–8 units/mL in digestion buffer (50 mM Tris [pH 8], 1 mM EDTA, 0.5% Triton X-100, 0.8 M guanidine HCl) and applied directly to gels in at least 10 times volume excess. The gels were then incubated in digestion buffer for at least 12 hr. Gels were then incubated with wash buffer (10% formamide, 2× SSC) for 2 hr at room temperature and hybridized with RNA FISH probes in hybridization buffer (10% formamide, 10% dextran sulfate, 2× SSC) overnight at 37°C. Following hybridization, samples were washed twice with wash buffer, 30 min per wash, and washed four times with water, 1 hr per wash, for expansion. Samples were imaged in water with 0.1 µg/mL DAPI.

Imaging cells were imaged using a Leica DMI600B automated widefield fluorescence microscope equipped with a X100 Plan Apo objective, a Pixis 1024BR cooled CCD (charge-coupled device) camera, a Prior Lumen 220 light source, and filter sets specific for each fluorophore. Images in each fluorescence channel were taken as a series of optical z-sections (0.3 µm per section).

Intron splice site strength (*Figure 2—figure supplement 3*) was calculated using MaxEnt (*Yeo and Burge, 2004*).

## Immunofluorescence

Staining for SC35 and polymerase II were performed with antibodies against SC35 (abcam ab11826, 1:200, NOTE: it has recently been described that this antibody may instead target SRRM2) and phospho S2 polymerase II (Active Motif, 61083, 1:200), respectively. Briefly, staining was performed on cells fixed and permeabilized as described above for RNA FISH. Primary antibody hybridization was carried out in 1× PBS overnight at 4°C. Samples were then washed with 1× PBS and incubated with secondary antibody (1:200) for 1 hr in 1× PBS at room temperature. Samples were then fixed for an additional 10 min in formaldehyde, washed with 1× PBS, and RNA FISH was performed as described above. RNA FISH data was quantified as described previously (*Johnson et al., 2000*). Briefly, cells were manually segmented, a Gaussian filter was applied to all spots, signal was distinguished from noise through semi-automated thresholding, each called spot was further fit to a Gaussian to get

sub-pixel resolution, and transcription sites were chosen based on a global brightness threshold (*Figure 1—figure supplement 1A and B*). Data was processed to assess distances and graphed in R. Calculation of mRNA length was based on the 75th percentile of 5'–3' distances of labeled mRNA seen in *Figure 1—figure supplement 2A*.

## Live-cell imaging

The live-cell imaging was conducted using human bronchial epithelial cells (single-cell clones containing 24× MS2 stem-loops in RAB7A first intron). To capture the diffusing RNA with MS2 labeling, live-cell imaging was performed on a Zeiss LSM780 laser scanning confocal microscope using 37°C incubation and 5% $CO_2$. Imaging was performed using 488 nm excitation, pinhole size of 2 airy units. The pixel size was 0.066 µm. One z-plane was imaged every 1 s for 60 frames. Transcription site intensity tracks were analyzed using KNIME pipeline (*Larson and Wan, 2019*). MSD analysis was performed using MATLAB.

Transcription site intensity tracks were then analyzed in MATLAB. Briefly, we identified the brightest pixel in each track, and called its location as the transcription site. We then cropped a 30×30 pixel region around this site, and used imresize() to increase resolution without interpolation. We then set a threshold of (((max pixel intensity – median pixel intensity)/2)+median pixel intensity) to create a binarized, Gaussian filtered image of the transcription site, for which we calculated the eccentricity using regionprops(). Images without clear, identifiable transcription sites were discarded.

## 4sU labeled chromatin-associated RNA sequencing and splicing index analysis

HeLa S3 cells (ATCC, CCL-2.2) were maintained in DMEM media containing 10% FBS, 100 U/mL penicillin, and 100 µg/mL streptomycin to 75% confluency. Cells were labeled in media containing 500 µM 4-thiouridine (4sU, Sigma, T4509) for 7.5 min. Plates were washed twice with 1× PBS and cells were lifted by scraping. Labeled cells were collected by centrifugation at 500×*g* for 2 min. To purify chromatin-associated RNA, steps 8–21 were followed exactly as described in *Smith et al., 1999*. In brief, nuclei were collected by lysing samples of 10 M cells in 200 µL cytoplasmic lysis buffer (0.15% [vol/vol] NP-40 [Thermo Fisher Scientific, 28324], 10 mM Tris-HCl [pH 7.0], and 150 mM NaCl) for 2 min, layering over a 500 µL sucrose cushion (10 mM Tris-HCl [pH 7.0], 150 mM NaCl, 25% [wt/vol] sucrose), and centrifuging at 16,000×*g* for 10 min. The nuclei pellet was washed in 800 µL wash buffer (0.1% [vol/vol] Triton X-100, 1 mM EDTA, in 1× PBS) and collected by centrifuging at 1150×*g* for 1 min. Nuclei were resuspended in 200 µL glycerol buffer (20 mM Tris-HCl [pH 8.0], 75 mM NaCl, 0.5 mM EDTA, 50% [vol/vol] glycerol, 0.85 mM DTT), and mixed with 200 µL nuclei lysis buffer (1% [vol/vol] NP-40, 20 mM HEPES [pH 7.5], 300 mM NaCl, 1 M urea, 0.2 mM EDTA, 1 mM DTT) by pulse vortex and incubated on ice for 2 min. The chromatin pellet was collected by centrifugation at 18,500×*g* for 2 min and resuspended in 1× PBS. All steps were performed at 4°C and buffers were prepared with 25 µM α-amanitin (Sigma, A2263), 0.05 U/µL SUPERase.In (Thermo Fisher Scientific, AM2694), and protease inhibitor mix (Roche, 11873580001). Chromatin-associated RNA was extracted using Qiazol lysis reagent (QIAGEN, 79306) following the manufacturer's instructions.

50 µg RNA per reaction was subjected to 4sU purification as described in *Girard et al., 2012*; *Wang et al., 1991*. In brief, labeled RNA (1 µg/10 µL) was incubated with 10% biotinylation buffer (100 mM Tris pH 7.5, 10 mM EDTA) and 20% EZ-Link Biotin-HPDP (1 mg/mL resuspended in DMF, Thermo Fisher Scientific, 21341) for 1.5 hr 24°C in the dark and 800 rpm to mix. RNA was purified by shaking the sample with a 1:1 volume of chloroform/isoamylacohol (24:1), separating using a phase-lock tube at 16,000×*g* for 5 min, and performing isopropanol precipitation. Biotinylated RNA was separated using the µMACS streptavidin kit (Miltenyi Biotec, 130-074-101) by mixing with µMACS streptavidin beads at a 2:1 ratio by volume at 800 rpm and 24°C for 15 min. RNA-streptavidin beads mix was transferred to the µMACS column and washed with wash buffer (100 mM Tris pH 7.5, 10 mM EDTA, 1 M NaCl, 0.1% Tween 20) at 65°C and room temperature three times each. Selected RNA was eluted off the magnet using the reducing agent, DTT (0.1 M), and purified using the miRNeasy micro kit (QIAGEN, 217084) with on-column DNase I treatment (QIAGEN, 79254). For the poly(A) depleted sample, the RNA was first concentrated using the RNA Clean and Concentrator kit (ZymoResearch, R1015). 10 µL Oligo(dT) Dynabeads (Thermo Fisher, 61002) were washed in 10 µL Binding Buffer (20 mM Tris.HCl pH 7.5, 1 M LiCl, and 2 mM EDTA). The sample was mixed with 10 µL binding buffer,

heated to 65°C for 2 min, moved to ice, and mixed with 1 µL SUPERase.In (Thermo Fisher Scientific, AM2694). The sample and beads were mixed thoroughly and annealed by rotating continuously on a mixer for 5 min at room temperature. Poly(A) RNA were collected on a magnet while depleted supernatant was removed and purified using the RNA Clean and Concentrator kit (ZymoResearch, R1015). Illumina sequencing libraries were prepared using the Ovation Universal RNA-seq System (NUGEN, 0343-32) with Universal Human rRNA strand selection reagent (NUGEN, S01859) following the manufacturer's instructions.

All samples were sequenced 2×80 on a NEXTseq 500 sequencer (Illumina, San Diego, CA, USA) in the Biopolymers Facility at Harvard Medical School. Paired-end reads were aligned to the ENSEMBLE GRCh38 (release 86) reference genome using STAR (v2.5.1a) (*Yeo and Burge, 2004*) with default parameters (except for readFilesCommand = cat, limitIObufferSize = 200000000, limitBAMsortRAM = 64000000000, outReadsUnmapped = Fastx, outSAMtype = BAM SortedByCoordinate, outSAMattributes = All, outFilterMultimapNmax = 101, outSJfilterOverhangMin = 3 1 1 1, outSJfilterDistToOtherSJmin = 0 0 0 0, alignIntronMin = 11, alignEndsType = EndToEnd). Splicing index calculations were determined by summing the number of spliced and unspliced read pairs that span exon junctions by at least three nucleotides and calculating the total spliced read pairs divided by the total unspliced read pairs for each gene; splicing index = 2 × spliced read pairs/(5'SS unspliced+3'SS unspliced read pairs).

Sequencing data is available on GEO under accession number GSE254859.

## Calculation of intron splicing order

For all independence calculations, it was first determined what proportion of RNA fell into each of the following categories:

1. Fully unspliced (exon spot with both introns)
2. Partially spliced with intron A removed
3. Partially spliced with intron B removed
4. Fully spliced (exon spot without either intron)

Spots within 0.65 µm of each other were considered to be colocalized spots (as described in *Figure 1*). Colocalized spots that contained exon, intron A, and intron B were designated as a 'three-color spot' (category 1). Spots that contained only exon signal were designated as fully spliced (category 4). Colocalized spots with only one intron signal were designated by the intron that was removed (e.g. if intron B is present, then intron A was removed, and the transcript is in category 2). Categories 2 and 3 represent mutually exclusive pathways to generate the final product of a mature transcript.

Once each transcript was categorized, we focused our attention on partially spliced transcripts, and determined the proportion of these transcripts in either category 2 or category 3. Should the splicing rate of different introns be remarkably different from each other, or should intron splicing be dependent on the splicing of other introns, we would expect an overrepresentation of one splicing pathway.

We labeled the fourth intron of TM4SF1 with two colored probes, and determined the number of intermediate transcripts identified using this method, allowing us to measure the noise inherent in this analysis method. With 100% detection efficiency, we would expect this set of two probes labeling the same intron to be spliced out of pre-mRNA simultaneously. To calculate a 'noise floor' of intermediate transcripts, we took the number of intermediate transcripts divided by two (as both category 2 and category 3 in this case measure the same type of error), and divided by the total number of transcripts captured in this experiment. Values below this noise floor are likely below our ability to faithfully detect differences in intron splicing behavior. Rows marked with ** in *Supplementary file 1B* show intron pairs that fall below this threshold, potentially due to the low number of intermediate transcripts captured for fast-splicing introns.

## Calculation of slow-moving zone dimension

To find the diameter of each (roughly circular) cloud, we used the equation $2 \cdot \sqrt{(median\,area)/\pi}$ , then converted these values from expanded space into unexpanded space by dividing by 4.65. Because the 5' end, 3' end, and intron clouds were of a similar size, we averaged the values from each cloud to give an approximate slow-moving zone diameter of 0.36 µm. Given the uncertainties inherent to

these calculations, these estimates are meant to provide an order of magnitude for the size of the zone, rather than precise measurements.

Below are the full calculations.

### 3' cloud

Median cloud area = 3.596 μm$^2$

Cloud diameter = $2 \cdot \sqrt{(3.596)/\pi}$ = 2.14 μm

In unexpanded space, cloud diameter = **0.46** μm

### 5' cloud

Median cloud area = 1.862 μm$^2$

Cloud diameter = $2 \cdot \sqrt{(1.862)/\pi}$ = 1.54 μm

In unexpanded space, cloud diameter = **0.33** μm

### Intron cloud

Median cloud area = 1.474 μm$^2$

Cloud diameter = $2 \cdot \sqrt{(1.474)/\pi}$ = 1.37 μm

In unexpanded space, cloud diameter = **0.29** μm

Approximate size of the slow moving zone: **0.36** μm

## Comparison to point source diffusion with degradation model

In order to evaluate whether diffusion of pre-mRNA from the site of transcription was compatible with our observations of clouds at the site of transcription revealed by expansion microscopy, we used a model of pre-mRNA diffusion that assumed a source at the site of transcription and a constant rate of degradation. The equation describing this model is given by *Femino et al., 1998*:

$$\frac{dC}{dt} = D\Delta C - \gamma C + R_0 \delta\left(r\right)$$

where $C$ is the concentration as a function of the radial distance and time, $D$ is the diffusion constant, $\gamma$ is the degradation rate, $R_0$ is the rate of production, and $\delta(r)$ is the delta function. The steady-state solution in 3D is given by:

$$C\left(r\right) = C_0 exp\left(\frac{-r}{\lambda}\right)$$

where $C_0$ is the concentration at the site of transcription, and $\lambda$ is sqrt($D/\gamma$).

We fit this formula to the radial distribution of pre-mRNA density from the region from 3 to 10 μm away from the transcription site, in which region we assume that the pre-mRNA are freely diffusing in the nucleoplasm. Note that our estimate for lambda in this region was roughly 3.34 μm. As a check for numerical consistency, we used an estimate of nucleoplasmic diffusion of 0.034 μm$^2$/s (*Coulon et al., 2014*) to obtain a degradation rate gamma of approximately 0.183 per minute, which yields an average lifetime of around 5.46 min for the pre-mRNA. This value is in line with what is generally accepted in the field and thus provided validation for the accuracy of our model and fit procedure. In order to see what concentration the diffusion and degradation model would predict at the transcription site itself, we extrapolated to the y-axis (radial distance = 0), yielding an estimated concentration of 18 molecules per cubic micron. We then measured the actual transcription site density at the site of transcription by counting the number of molecules in the vicinity using expansion microscopy and dividing by an estimated transcription site volume, yielding 235 molecules per cubic micron. (The volume was estimated by taking the area of the transcription site spots in 2D, estimating the radius of a putatively circular transcription site, and then using that radius to compute the volume of a 3D sphere.)

## Acknowledgements

We would like to thank members of the Raj, Churchman, Phillips-Cremins, and Berger labs for critical reading of the manuscript. We thank Hyun Youk for discussions about radial distributions. AR and JPC acknowledge support from NIH 4DN U01 HL129998, NIH 4DN U01DK127405, and NSF EFMA19334000. AR additionally acknowledges NIH R01 CA238237, NIH Director's Transformative Research Award R01 GM137425, NIH R01 CA232256, NSF CAREER 1350601, NIH P30 CA016520, NCI SPORE P50 CA174523, NIH U01 CA227550, NIH Center for Photogenomics (RM1 HG007743), and the Tara Miller Foundation. AC was also supported by NIH training grant T32 GM-07229. AO acknowledges support from the NSF-GRFP. LSC acknowledges support from NIH R21-HG009264 and NIH R01-GM117333, and NIH F31-GM122133 to HLD. SB acknowledges support from NIH 3R01CA078831-20S1 and KAA acknowledges support from NIH 5F32CA221010-02.

## Additional information

### Funding

| Funder | Grant reference number | Author |
|---|---|---|
| National Institutes of Health | T32 GM-07229 | Allison Coté |
| National Science Foundation | NSF-GRFP | Aoife O'Farrell |
| National Institutes of Health | R21-HG009264 | L Stirling Churchman |
| National Institutes of Health | R01-GM117333 | L Stirling Churchman |
| National Institutes of Health | F31-GM122133 | Heather L Drexler |
| National Institutes of Health | 3R01CA078831-20S1 | Sareh Bayatpour |
| National Institutes of Health | 5F32CA221010-02 | Katherine A Alexander |
| National Institutes of Health | U01 HL129998 | Arjun Raj Jennifer Phillips-Cremins |
| National Institutes of Health | U01DK127405 | Arjun Raj Jennifer Phillips-Cremins |
| National Science Foundation | EFMA19334000 | Arjun Raj Jennifer Phillips-Cremins |
| National Institutes of Health | R01 CA238237 | Arjun Raj |
| National Institutes of Health | R01 GM137425 | Arjun Raj |
| National Institutes of Health | R01 CA232256 | Arjun Raj |
| National Science Foundation | 1350601 | Arjun Raj |
| National Institutes of Health | P30 CA016520 | Arjun Raj |
| National Cancer Institute | P50 CA174523 | Arjun Raj |
| National Institutes of Health | U01 CA227550 | Arjun Raj |
| National Human Genome Research Institute | RM1 HG007743 | Arjun Raj |

| Funder | Grant reference number | Author |
|---|---|---|
| Tara Miller Foundation | | Arjun Raj |

The funders had no role in study design, data collection and interpretation, or the decision to submit the work for publication.

## Author contributions

Allison Coté, Conceptualization, Data curation, Software, Formal analysis, Validation, Investigation, Visualization, Methodology, Writing – original draft; Aoife O'Farrell, Conceptualization, Data curation, Software, Formal analysis, Validation, Investigation, Visualization, Methodology, Writing – original draft, Writing – review and editing; Ian Dardani, Margaret Dunagin, Chris Coté, Yihan Wan, Sareh Bayatpour, Heather L Drexler, Katherine A Alexander, Fei Chen, Asmamaw T Wassie, Rohan Patel, Kenneth Pham, Investigation, Methodology; Edward S Boyden, Shelly Berger, Jennifer Phillips-Cremins, L Stirling Churchman, Resources, Supervision, Funding acquisition; Arjun Raj, Conceptualization, Resources, Supervision, Funding acquisition, Writing – original draft, Project administration, Writing – review and editing

## Author ORCIDs

Allison Coté ⓘ http://orcid.org/0000-0001-9567-3184
Aoife O'Farrell ⓘ http://orcid.org/0009-0007-3091-2608
Ian Dardani ⓘ http://orcid.org/0000-0002-0335-7422
Margaret Dunagin ⓘ http://orcid.org/0000-0002-8344-2570
L Stirling Churchman ⓘ http://orcid.org/0000-0003-3888-2574
Arjun Raj ⓘ http://orcid.org/0000-0002-2915-6960

Reviewer #1 (Public Review): https://doi.org/10.7554/eLife.91357.3.sa1
Reviewer #2 (Public Review): https://doi.org/10.7554/eLife.91357.3.sa2
Author response https://doi.org/10.7554/eLife.91357.3.sa3

# Additional files

## Supplementary files

• MDAR checklist

• Supplementary file 1. Tables showcasing distal post-transcriptional splicing and splicing order. (A) Dispersal and percent of splicing that occurs distally post-transcriptional, as assayed through RNA FISH. Each row indicates one probe set. (B) Percent of intermediate splicing products that splice in one order versus the other, indicating that splicing can generally occur in any order.

• Supplementary file 2. List of oligo sequences used to generate FISH probes.

## Data availability

All data and code to generate figures can be found on the Dryad repository https://doi.org/10.5061/dryad.m0cfxppb7. Sequencing data is available on GEO under accession number GSE254859.

The following datasets were generated:

| Author(s) | Year | Dataset title | Dataset URL | Database and Identifier |
|---|---|---|---|---|
| Coté A, Drexler H, Churchman LS, Raj A | 2024 | Post-transcriptional splicing can occur in a slow-moving zone around the gene | https://www.ncbi.nlm.nih.gov/geo/query/acc.cgi?acc=GSE254859 | NCBI Gene Expression Omnibus, GSE254859 |
| Coté A, Drexler H, Churchman LS, Raj A | 2024 | Post-transcriptional splicing can occur in a slow-moving zone around the gene | https://doi.org/10.5061/dryad.m0cfxppb7 | Dryad Digital Repository, 10.5061/dryad.m0cfxppb7 |

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
