## [Editor Report · eLife assessment]

This **fundamental** study addresses a long-standing mystery in splicing regulation: does splicing occur co- or post-transcriptionally? The authors provide **compelling** evidence demonstrating that splicing can occur post-transcriptionally at a transcription site proximal zone, changing the way we think about splicing.

---

## [Referee Report · Reviewer #1 (Public Review)]

The manuscript has helped address a long-standing mystery in splicing regulation: whether splicing occurs co- or post-transcriptionally. Specifically, the authors (1) uniquely combined smFISH, expansion microscopy, and live cell imaging; (2) revealed the ordering and spatial distribution of splicing steps; and (3) discovered that nascent, not-yet-spliced transcripts move more slowly around the transcription site and undergo splicing as they move through the clouds. Based on the experimental results, the authors suggest that the observation of co-transcriptional splicing in previous literature could be due to the limitation of imaging resolution, meaning that the observed co-transcriptional splicing might actually be post-transcriptional splicing occurring in proximity to the transcription site. Overall, the work presented here clearly provides a comprehensive picture of splicing regulation.

---

## [Referee Report · Reviewer #2 (Public Review)]

Allison Coté et al. investigated the ordering and spatial distribution of nascent transcripts in several cells using smFISH, expansion microscopy, and live-cell imaging. They find that pre-mRNA splicing occurs post-transcriptionally at the clouds around the transcription start site, termed the transcription site proximal zone. They show that pre-mRNA may undergo continuous splicing when they pass through the zone after transcription. These data suggest a unifying model for explaining previously reported co-transcriptional splicing events and provide a direction for further study of the nature of the slow-moving zone around the transcription start site.

This paper is well-written. The findings are very important, and the data supports the conclusions well. However, some aspects of the image and description need to be clarified and revised.

1. The sentence "By distinguishing the separate fluorescent signals from probes bound to exons and introns, we could visualize splicing intermediates (represented by colocalized intron and exon spots) relative to the site of transcription (represented by bright colocalized intron and exon spots) and fully spliced products (represented by exon spots alone)." is accidentally repeated twice, one of them should be deleted.

2. The authors describe Figure 4E and 4F results in the main text as that "we performed RNA FISH simultaneously with immunofluorescence for SC35, a component of speckles, and saw that these compartmentalized pre-mRNA did indeed appear near nuclear speckles both before (Supplementary Figure 6C) and after (Figure 4E) splicing inhibition." However, no SC35 staining is shown in the Figure 4E. A similar situation happened in describing Figure 4F.

---

## [Author Response]

The following is the authors’ response to the original reviews.

**Reviewer #1 (Public Review):**
The manuscript has helped address a long-standing mystery in splicing regulation:whether splicing occurs co- or post-transcriptionally. Specifically, the authors (1) uniquely combined smFISH, expansion microscopy, and live cell imaging; (2) revealed the ordering and spatial distribution of splicing steps; and (3) discovered that nascent, not-yet-spliced transcripts move more slowly around the transcription site and undergo splicing as they move through the clouds. Based on the experimental results, the authors suggest that the observation of co-transcriptional splicing in previous literature could be due to the limitation of imaging resolution, meaning that the observed co-transcriptional splicing might actually be post-transcriptional splicing occurring in proximity to the transcription site. Overall, the work presented here clearly provides a comprehensive picture of splicing regulation.Major points:1. Linearity of expansion microscopy. For Figure 2B, it would be helpful to display the same sample before and after expansion, just like Supplementary Figure 3, but with a transcription site and "cloud". In the current version, the transcription site looks quite different in the not-expanded (more green dots on the left) and expanded image (more green dots on the top).

We thank the reviewer for this comment on linearity of expansion. Based on our prior manuscript (Chen et al 2015 Nature Methods. PMID: 27376770), we expect expansion microscopy to yield isotropic expansion. Indeed, as shown in Figure 2 Figure Supplement 1, we confirmed that expansion of nuclei (1B, top) and transcripts (1B, bottom) is isotropic. Additionally, before splicing inhibition, we demonstrated the linearity of expansion for a transcription site (1B, left), shown at standard resolution with intron stain. The images shown in Figure 2B are meant solely to illustrate the change in resolution upon expansion, and are not meant to imply spatial matching between the expanded and unexpanded image. We apologize for the confusion and have clarified this in the figure legend for Figure 2.

We also point the reader towards Figure 2 Figure Supplement 2, in which we validate the use of expansion microscopy in these findings. We show that transcription sites in expanded samples were the same size as those imaged using stochastic optical reconstruction microscopy (STORM), demonstrating that expansion did not significantly alter the morphology of the site.

1. FISH dot colocalization. What is the colocalization rate of FISH dots in general under experimental conditions? In addition, in Figures 2C and 2G, why do some 3'exon dots not have co-localized 5'exon dots?

We thank the reviewer for asking for these important clarifications. Under standard (non-expanded) conditions, our colocalization of 3’ and 5’ spots varies by gene, but more than 75% of intron spots colocalize with exon spots for the vast majority of transcripts we evaluated. The percentage of colocalization for each gene and intron can be found in column 4 of Supplementary File 1A.

Regarding the second point—these individual images may not reflect the actual quantitative number of spot counts at the site, as these transcription sites have a sizable Z dimension that is difficult to capture in one image, and certain dyes are more easily visually distinguished in contrasted images than others. These factors may cause some 3’ spots to appear without a corresponding co-localized 5’ spot in these images. We refer the reviewer to Figure 2 Figure Supplement 2C for quantitative spot counting of an expanded transcription site, for which there are a similar number of 3’ end and 5’ end spots within the entire Z-stacked image. Importantly, these transcription site clouds contain longer, unspliced transcripts, potentially leading to further separation between the 5’ and 3’ ends of a single transcript when compared to a cytoplasmic, spliced transcript (quantified in Figure 2I).

1. It would be helpful if the authors uploaded a few examples of live cell imaging movies.

Certainly! Please refer to the new Videos 1-3 for representative examples of live cell imaging data.

1. It is recommended to double-check the text for errors.

We apologize for errors in the original manuscript, and have made the appropriate corrections.

**Reviewer #2 (Public Review):**
Allison Coté et al. investigated the ordering and spatial distribution of nascent transcripts in several cells using smFISH, expansion microscopy, and live-cell imaging. They find that pre-mRNA splicing occurs post-transcriptionally at the clouds around the transcription start site, termed the transcription site proximal zone. They show that pre-mRNA may undergo continuous splicing when they pass through the zone after transcription. These data suggest a unifying model for explaining previously reported co-transcriptional splicing events and provide a direction for further study of the nature of the slow-moving zone around the transcription start site.This paper is well-written. The findings are very important, and the data supports the conclusions well. However, some aspects of the image and description need to be clarified and revised.The authors describe Figure 4E and 4F results in the main text as that "we performedRNA FISH simultaneously with immunofluorescence for SC35, a component of speckles, and saw that this compartmentalized pre-mRNA did indeed appear near nuclear speckles both before (Supplementary Figure 6C) and after (Figure 4E) splicing inhibition." However, no SC35 staining is shown in the Figure 4E. A similar situation happened in describing Figure 4F.

We thank the reviewer for noting this error. We mistakenly called in text for Figure 4E, when we meant to refer to Figure 4G, which shows combined RNA FISH and SC35 immunofluorescence show compartmentalization within nuclear speckles. Figures 4E and 4F do not show SC35 immunofluorescence. We have altered the text and figure captions accordingly.

Minor points:1. For Figures, it would be better to mark co-transcriptional and proximal post-transcriptional splicing in a clearer way. Like in Figure 1A, the simulated RNA FISH signals are almost identical across two conditions, which is a bit confusing. Overlapping and close proximity shall be better illustrated in related figures.

We thank the reviewer for these suggestions. We have iterated these figures through multiple revisions and have found that these diagrams tend to resonate the most, so we have elected to keep them as is, but we do appreciate the suggestion.

1. May include some details of expansion microscopy in the last paragraph of the Introduction. For example, why introduce expansion microscopy? To what level it can help overcome the diffraction limit?

We thank the reviewer for this comment, and have added additional text to this paragraph to further set up the use of expansion microscopy.

1. Double-check the formatting. Some sub-titles are in Bold, some in Italic.

We apologize for any formatting errors, and have made the appropriate corrections.

1. Please double-check the writing. I find many incompatible parts across the manuscript. For example, as described in the Figure 1D caption, there aren't "first" and "second" graphs in the figure. Moreover, some writings require additional refinement. For instance, in the Introduction part, the paragraph discussing RNA imaging, various techniques (such as FISH and live imaging), and concerns (such as microscopy resolution, chromatin fraction, and limitations related to reporter genes) are intertwined without clear indexing or logical structuring. Similar cases in other paragraphs too. Last but not least, I can even find repetitive sentences across the manuscript. For instance, I believe that the authors forgot to delete "By distinguishing the separate fluorescent signals from probes bound to exons and introns, we could visualize splicing intermediates (represented by colocalized intron and exon spots) relative to the site of transcription (represented by bright colocalized intron and exon spots) and fully spliced products (represented by exon spots alone)." in the first paragraph of the Results part, as the exact same sentence re-occurs right after. I've only listed a few examples here. Please refine the manuscript.

We apologize for any errors in the original manuscript, and have made the appropriate corrections.

**Reviewer #2 (Recommendations For The Authors):**
1. The sentence "By distinguishing the separate fluorescent signals from probes bound to exons and introns, we could visualize splicing intermediates (represented by colocalized intron and exon spots) relative to the site of transcription (represented by bright colocalized intron and exon spots) and fully spliced products (represented by exon spots alone)." is accidentally repeated twice, one of them should be deleted.

We apologize for this duplication, and have made the appropriate correction.